# An initiative for a more inclusive working life and its effect on return-to-work after sickness absence: a multistate longitudinal cohort study

Rune Hoff [ID],[1] Niklas Maltzahn,[1,2] Rachel Louise Hasting,[3] Suzanne L Merkus,[4] Karina Undem,[3] Petter Kristensen [ID],[3] Ingrid Sivesind Mehlum,[3,5] Jon Michael Gran[1,2]

**Correspondence to**
Dr Rune Hoff;
rune.hoff@medisin.uio.no

## ABSTRACT

**Objectives** To reduce sickness absence (SA) and increase work participation, the tripartite Agreement for a More Inclusive Working Life (IA) was established in Norway in 2001. IA companies have had access to several measures to prevent and reduce SA. Our aim in this paper was to estimate the average effect of having access to IA at the time of entering a first SA on later return-to-work (RTW) and on time spent in other work-related states. A secondary objective was to study how effects varied between women and men, and individuals with SA due to either musculoskeletal or psychological diagnoses.

**Design** Population-based observational multistate longitudinal cohort study.

**Setting** Individual characteristics and detailed longitudinal records of SA, work and education between 1997-2011 were obtained from population-wide registries.

**Participants** Each individual born in Norway 1967–1976 who entered full-time SA during 2004–2011, with limited earlier SA, was included (n=187 930).

**Primary and secondary outcome measures** Individual multistate histories containing dated periods of work, graded SA, full-time SA, non-employment and education.

**Methods** Data were analysed in a multistate model with 500 days of follow-up. The effect of IA was assessed by estimating differences in state probabilities over time, adjusted for confounders, using inverse probability weighting.

**Results** IA increased the probability of work after SA, with the largest difference between groups after 29 days (3.4 percentage points higher (95% CI 2.5 to 4.3)). Differences in 1-year expected length of stay were 8.4 additional days (4.9 to 11.9) in work, 7.6 (4.8 to 10.3) fewer days in full-time SA and 1.6 (-0.2 to 3.4) fewer days in non-employment. Similar trends were found within subgroups by sex, musculoskeletal and psychological diagnoses. The robustness of the findings was studied in sensitivity analyses.

**Conclusion** Measures to prevent and reduce SA, as given through IA, were found to improve individuals' RTW after entering SA.

## STRENGTHS AND LIMITATIONS OF THIS STUDY

⇒ This is the first study on the effects of the Norwegian population-wide initiative to increase work-participation that analyses a large dataset of longitudinal multistate outcomes.
⇒ The data exhibit a high level of completeness and include also detailed covariate information.
⇒ Limitations of the study are that we only had access to observational data and that Agreement for a More Inclusive Working Life was not randomised.

## INTRODUCTION

Sickness absence (SA) from work is an important long-term health outcome for the individual, as well as a social and financial burden for society.[1–3] It has been shown that while in SA, individuals are at greater risk of a more permanent exit from working life.[4 5] SA increased greatly in Norway between 1995 and 2000, mainly due to an increase in long-term SA (more than 16 calendar days). The yearly average days of SA per worker in private workplaces, increased from 8.8 days to 12.9 days in the same period.[6] It should be noted that this increase came after a period of reduction in SA,[7] and that around one-third of the increase can be explained by changes in the size and age of the workforce.[6]

In 2001, the Norwegian Government reached out to major partners in working life, representing both workers and employers, with the intention of reducing SA and prolonging working life. This resulted in the tripartite Agreement for a More Inclusive Working Life (IA), which is still active, and included three operative goals: (1) reduce SA by at least 20 %, (2) increase employment for individuals with functional limitations and (3) increase the average retirement age.[6] By 2004, 55% of workers were in IA companies,

that is, companies that signed a cooperation agreement with the Norwegian Labour and Welfare Administration (NAV) and the employee representatives in the company, to cooperate systematically to achieve more inclusive workplaces.[8]

IA companies and their employees have access to several measures to reduce and prevent SA. These include special regulations and support from working life centres that were established regionally by NAV. The support from these centres can be comprehensive, including tailored support for employees with health problems.[9] The measures included the possibility for employees in SA to use so-called active sickness benefits, where they may return-to-work (RTW) and perform modified tasks while still receiving payments from NAV. Employees in IA companies are also allowed to have longer self-certified SA (up to eight consecutive calendar days, instead of three, and a total of 24 annual days, instead of 12). The IA companies may receive refunds for measures related to aiding employees in returning to work. If it is impossible for an employee to return to the same position they had before SA, the IA companies have agreed to cooperate with NAV in assisting in retraining so that the employee can continue to work in the same company.

Since implemented, IA has not met the goals regarding SA, and systematic evaluations, although sparse, suggest limited or no effect on the risk of experiencing SA.[8 10–12] Studies of other general workplace interventions have found that they were generally not effective in reducing SA.[13] However, there are studies suggesting effects of workplace interventions on RTW.[12 14]

The main objective of this study was to estimate the average effect of having access to IA at the time of entering a first SA on later RTW and on time spent in other work-related states for the following 500 days. To analyse such individual outcome trajectories in more detail than have been done before, we used a multistate model,[15] capturing individuals' movement between the states of full-time SA, graded SA, work, non-employment and education over time. Inverse probability of treatment (IPT) weighting was used to adjust for confounding between IA and multistate outcomes.[16] The effect of IA has not previously been studied in detail at a population level using such methods. A secondary objective was to study how effects varied between women and men, and individuals with SA due to either musculoskeletal or psychological diagnoses.

## METHODS
### Design, participants and exposure
To meet the study objectives, we used observational longitudinal registry data from a cohort consisting of all individuals born in Norway between 1967 and 1976 (626 928 individuals), as registered by the Medical Birth Registry of Norway. The same cohort is described in greater detail by Kristensen and Bjerkedal.[17] Given limited earlier SA (no SA in 2003), individuals were included in the analysis

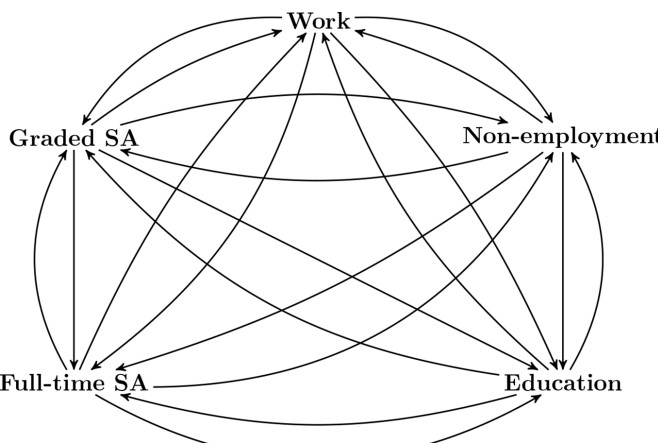

**Figure 1** The multistate model showing possible transitions after individuals start in in full-time sickness absence (SA). Individuals may move repeatedly between full-time SA, graded SA, work, non-employment and education. Note that death was a competing transition in all states, but left out of the figure.

if they entered full-time SA (>16 calendar days, counted from the first day), between 1 January 2004, when IA was properly implemented, and 31 December 2010. In total, this comprised 187 930 persons. Baselines were set to the dates of the initial SA episode for all individuals, who simultaneously had their IA exposure (yes/no) recorded, based on the IA status of the company they worked for, and were followed for 500 days or until death, right-censoring or administrative censoring on 1 January 2011. The choice of 500 days was a pragmatic choice, meant to cover the first full year and a short period beyond. The immediate period after the first year is of interest, as welfare policies impose a change in benefits received after a full year of receiving sick leave benefits. Moreover, for even longer follow-up periods, individuals' IA status would be more likely to change, making results from an analysis based on baseline IA status more difficult to interpret.

### Patient and public involvement
Patients or the public were not involved in the design, or conduct, or reporting, or dissemination plans of our research.

### Outcomes and confounders
The individual multistate histories are described by dated transitions between periods of work, graded SA, full-time SA, non-employment, education and death, referred to as states, as shown in figure 1. Multistate models can be seen as extensions of standard time-to-event models with only two states, such as alive and dead. In multistate models, individuals are allowed to have several transitions while remaining under observation. In our application, individuals started in full-time SA and could move back and forth between transient states at any time during the 500 days follow-up period, and for example have multiple spells of full-time SA. The exception is death, which is a

so-called absorbing state, where further transitions are not possible.

As the IA Agreement is not randomly assigned to individuals, the association between IA and the individual multistate histories may be confounded by various characteristics that are different between individuals and workplaces with and without IA and need to be adjusted for. For this purpose, information was collected at individual level on the assumed confounders recorded at individuals' baseline: age, sex, education level, calendar year, company size, geographical region and type of industry. Education level was the individual's highest attained degree, separated into four categories: lower secondary, upper secondary, undergraduate and graduate. Type of industry covered 20 categories in the Statistical Classification of Economic Activities in the European Community.[18] Company size was based on yearly records of number of employees, and company region comprised southern, eastern, western, middle and northern parts of Norway.

### Data sources and data preparation

All the data were collected from Statistics Norway's event database of employment and welfare (FD-Trygd), The National Education Database and the registers of the NAV. The data contained individual and company characteristics acting as potential baseline confounders, individual status of the main intervention (IA status), dated records of all individuals' work (with taxable income), SA (>16 calendar days) and education. From the records of work, SA and education, we generated individual multistate outcome histories, or trajectories, over a period of 500 days, with the possible states of full-time SA, graded SA, work, education and non-employment.

Individuals were included in the study on the date of their first full-time SA episode after 1 January 2004, given that they had not been in SA the previous year. All the baseline confounders were set to the values as recorded on each individual inclusion date. IA status was set to the IA status of the individual's company at the time of inclusion. Individuals working for more than one company during follow-up were seen as having IA if either of the workplaces had IA. For this study, SA included periods eligible for either of three different types of benefits in the Norwegian welfare system: sick leave benefit, work assessment allowance (medical or vocational rehabilitation benefits) and disability pension, all attainable as full-time (100%) or graded (<100%). Work referred to paid employment and included holidays and parental leave, while education covered admittance to officially licensed education programmes. Periods without SA, work or education were considered as non-employment if they had later records of SA, work or education. Lost to follow-up without any later records were treated as right-censoring. Whenever different types of records occurred simultaneously on the same dates for an individual, different records had precedence in the following order, from most to least important: SA, work and lastly education.

### Subgroup and sensitivity analyses

In addition to the main analysis, separate analyses were done for women and men, and for diagnosis group associated with initial SA (musculoskeletal or psychological). Musculoskeletal diagnoses consisted of ICPC-2 (International Classification of Primary Care)[19] codes starting with L, while psychological diagnoses referred to codes starting with P.

We conducted five sensitivity analyses. First, we compared results adjusting for subsets of the confounders, to see how omitted variables in the adjustment set could bias results. In the second, we adjusted for additional potential confounders available for a male subgroup of the study population, describing IQ, physical stamina, BMI and eligibility for military service. In the third, we used pre-baseline longitudinal outcomes between 1997 and 1999, as negative outcome controls.[20] If confounding was sufficiently adjusted for, state transition histories before baseline should be similar across IA groups, after weighting with the same IPT weights as in the main analysis. In the fourth, we analysed a subset of industries with mostly private companies. Lastly, since our study population were in the age span 28–43 at entry into the study, and a substantial part (20,355) of the initial SA cases was in connection with pregnancy, childbearing and family planning (ICPC-2 codes starting with W), an analysis where these cases were removed was conducted.

Although type of industry is adjusted for in the main analysis, we also created sequence plots of individuals' multistate histories within five different industry sectors for descriptive purposes. These plots illustrate typical patterns of multistate trajectories at the individual level, over the 500 days of follow-up. We also looked at gender balance within the largest industries of our study population for discussion purposes.

A closer description of these analyses is found in online supplemental material 1.

### Estimands and statistical methods

The individual multistate outcomes were analysed using hazard based multistate models for time-to-event data,[15] by first fitting hazard models for each transition and then estimating state probabilities for all states in figure 1. The model allows individuals to move, possibly back and forth, between states over time according to the possible transitions illustrated in the figure. By state probabilities, we here mean the probability of being in a particular state at a given time-point, which can be seen as an extension of regular survival probabilities (eg, Kaplan-Meier curves). To assess the effect of IA, we sought to identify the average treatment effect, as the difference in state probabilities $\theta(t) = \pi^1(t) - \pi^0(t)$, where $\pi^a(t)$ is the adjusted marginal state probability, corresponding to the state probability we would have seen if everyone in the target population was fixed to having IA status equal to $a$ (taking the values 1 and 0 for IA and no IA, respectively). From the state probabilities, we also calculated the expected length of

stay (ELOS)[21] in the different states over the first year, for the situations where everyone was fixed to $a=1$ or $a=0$.

Confounding adjustment was done by calculating baseline stabilised IPT weights,[22] also known as propensity score weights, and then fitting weighted additive hazards regression models[23] for every possible transition in figure 1, adjusting only for IA status. This corresponds to using weighted Nelson-Aalen estimators for the cumulative hazards separately for the two IA groups. The corresponding state probabilities were then estimated by plugging the transition-specific cumulative hazard estimates into the Aalen-Johansen estimator. The ELOS can be calculated directly from the Aalen-Johansen estimates, as the area under the curve of the corresponding state probabilities over time. To calculate the IPT weights, we modelled IA status as a function of the baseline confounders in a logistic regression model. Since IA is an intervention at the company level, we calculated 95% CIs using a clustered bootstrap where the resampling is done on company level.[24]

For a more formal specification of the target estimands and details on the statistical methods, see online supplemental material 1, and, more generally, Andersen and Keiding,[15] Gran et al[16] and Aalen et al.[23] All analyses were performed in R, on the Services for Sensitive Data facilities, owned by the University of Oslo. R code is available on GitHub (see the Declarations section).

## RESULTS
### Descriptive statistics

Over a total follow-up of 90 231 405 person days, the 187 930 individuals included made a total of 457 610 transitions, which are summarised in detail in online supplemental table 1. A summary of the individual characteristics can be found in table 1. In the second and third column of the latter table, we can see how the measured confounders are distributed differently among the IA and non-IA individuals, with, for example, more women and individuals in larger workplaces having IA. In the fourth and fifth column, however, we see how the measured confounders are distributed in the weighted data, after applying the IPT weights, with the measured confounding characteristics now being well balanced between groups.

First, we fitted a multistate model without covariates, to calculate the observed state probabilities in the entire cohort, regardless of IA status. A stacked probability plot is shown in figure 2. All individuals started in full-time SA and could thereafter move to other states. The proportion of individuals who had returned to work reached 50.6% (95% CI 50.4% to 50.8%) after 50 days, and 64.7% (95% CI 64.5% to 64.9%) after 100 days. Just before passing 1 year, 77.0% (95% CI 76.8% to 77.2%) were in work, 3.3% (95% CI 3.2% to 3.4%) were in graded SA and 9.9% (95% CI 9.8% to 10.0%) in full-time SA. At the same time, 9.2% (95% CI 9.0% to 9.3%) were in non-employment and 0.5% (95% CI 0.47% to 0.53%) in education. After 1 year, we see distinct shifts in the probabilities, coinciding

with maximum allowed days of consecutive sick leave benefit. After this, individuals either RTW or typically receive other welfare benefits, including work assessment allowance.

### Main analysis

To assess the effect of IA versus no IA, we compared state probabilities in an IPT weighted multistate model. A summary of individual characteristics in the weighted population can be found in table 1. Figure 3 shows the absolute difference in state probabilities for IA versus no IA over the follow-up period, with 95% CI's based on 1000 bootstrap samples. In the figures, death was left out, as differences here were negligible. The figure shows how IA increased the probability of RTW and decreased the probability of full-time SA in the period following an initial SA episode. Peak effect of IA on RTW occurred after 1 month, where the probability was about 3.4 percentage points (pp) higher with IA.

The difference declined to below 2 pp after one year. The effect on full-time SA followed an opposite pattern for the first 2–3 months and declined to zero after 1 year. The effect on graded SA was small (increase), but the confidence intervals included zero. The effect on non-employment was small initially, but amounted to around 1 pp reduction after 1 year.

The average individual effects of IA versus no IA are summarised in terms of ELOS in table 2. Here, we see that the estimated ELOS difference between IA and no IA for the first year was 8.4 more days (95% CI 4.9 to 11.9 days) in work, 7.6 (95% CI 4.8 to 10.3) fewer days in full-time SA and 1.6 (95% CI −0.2 to 3.4) fewer days in non-employment. Differences for graded SA and education were smaller.

### Subgroup analyses

Results from the subgroup analyses are plotted in figure 3. Similar effect patterns as before were found, however, effects of IA were higher for men than for women. Effects were also higher in both selected diagnosis groups compared with in the overall analysis, although only slightly for musculoskeletal diagnoses.

For women, the effect of IA on RTW from full-time SA, reached a peak after 25 days, where the probability was 2.8 pp (95% CI 1.9 to 3.7) higher with IA. The effect remained close to 2 pp higher for roughly 200 days, before declining to around one pp 1 year after inclusion. After 1 year, the effect was close to zero. The effect on work was almost entirely offset by reduced probability of full-time SA, while effects on other states were smaller. The peak effect for men on RTW were 29 days after starting full-time SA, with 4.5 pp (3.0 to 6.0) higher probability for men with IA. Contrary to in women, the effect on work stayed positive, never dropping below 2 pp, the entire follow-up period. The effect on work was mostly offset by reduced probability of full-time SA, but with more noticeable effects on other states, compared with women. There was a small increase in graded SA (after initially starting

**Table 1** Descriptive statistics for individuals working in companies with the Inclusive working life Agreement (IA), or without, at the time of inclusion. The table also describes the inverse probability of treatment weighted population.

| | No IA, n (%) | IA, n (%) | No IA (IPTW) n (%) | IA (IPTW) n (%) |
|---|---|---|---|---|
| **Sex** | | | | |
| Women | 57 002 (0.48) | 45 139 (0.66) | 63 383 (0.53) | 37 907 (0.55) |
| Men | 62 361 (0.52) | 23 428 (0.34) | 55 037 (0.46) | 30 885 (0.45) |
| **Age at inclusion** | | | | |
| 28–32 | 33 876 (0.28) | 16 336 (0.24) | 31 722 (0.27) | 18 619 (0.27) |
| 33–37 | 60 561 (0.51) | 34 485 (0.50) | 60 150 (0.50) | 35 121 (0.51) |
| 38–43 | 24 926 (0.21) | 17 746 (0.26) | 26 547 (0.22) | 15 052 (0.22) |
| **Inclusion year** | | | | |
| 2004 | 31 255 (0.26) | 12 762 (0.19) | 28 556 (0.24) | 17 376 (0.25) |
| 2005 | 25 935 (0.22) | 12 671 (0.18) | 24 741 (0.21) | 14 278 (0.21) |
| 2006 | 19 812 (0.17) | 11 373 (0.17) | 19 757 (0.17) | 11 169 (0.16) |
| 2007 | 14 951 (0.13) | 9895 (0.14) | 15 307 (0.13) | 8769 (0.13) |
| 2008 | 11 955 (0.10) | 9094 (0.13) | 12 704 (0.11) | 7258 (0.11) |
| 2009 | 10 082 (0.08) | 8029 (0.12) | 11 234 (0.09) | 6366 (0.09) |
| 2010 | 5373 (0.05) | 4743 (0.07) | 6121 (0.05) | 3578 (0.05) |
| **Education** | | | | |
| Lower secondary | 39 162 (0.33) | 17 005 (0.25) | 35 319 (0.30) | 19 753 (0.29) |
| Upper secondary | 52 977 (0.44) | 24 137 (0.35) | 48 767 (0.41) | 28 885 (0.42) |
| College | 23 393 (0.20) | 23 754 (0.35) | 28 851 (0.24) | 17 392 (0.25) |
| University | 3831 (0.03) | 3671 (0.05) | 5483 (0.05) | 2763 (0.04) |
| **Industry** | | | | |
| Agriculture/forestry | 1406 (0.01) | 106 (0.00) | 963 (0.01) | 720 (0.01) |
| Commercial services | 7015 (0.06) | 1811 (0.03) | 5633 (0.05) | 3381 (0.05) |
| Construction | 12 357 (0.10) | 2763 (0.04) | 9571 (0.08) | 4785 (0.07) |
| Education | 3959 (0.03) | 9624 (0.14) | 9594 (0.08) | 4941 (0.07) |
| Scientific service work | 5475 (0.05) | 853 (0.01) | 4006 (0.03) | 2120 (0.03) |
| Electricity supply | 387 (0.00) | 317 (0.00) | 473 (0.00) | 289 (0.00) |
| Entertainment | 1502 (0.01) | 379 (0.01) | 1203 (0.01) | 944 (0.01) |
| Financial | 1701 (0.01) | 1189 (0.02) | 1923 (0.02) | 1282 (0.02) |
| Health and social work | 16 658 (0.14) | 27 779 (0.41) | 25 274 (0.21) | 15 954 (0.23) |
| Hotel and restaurant | 3873 (0.03) | 856 (0.01) | 2994 (0.03) | 1543 (0.02) |
| Information/communication | 5087 (0.04) | 1432 (0.02) | 4126 (0.03) | 2228 (0.03) |
| Manufacturing | 14 306 (0.12) | 7560 (0.11) | 13 694 (0.11) | 7429 (0.11) |
| Mining/quarrying | 2584 (0.02) | 1075 (0.02) | 2331 (0.02) | 1394 (0.02) |
| Other service | 2166 (0.02) | 554 (0.01) | 1731 (0.01) | 1007 (0.01) |
| Public administration | 3765 (0.03) | 5528 (0.08) | 6850 (0.06) | 3756 (0.05) |
| Real estate | 1116 (0.01) | 90 (0.00) | 768 (0.01) | 493 (0.01) |
| Transport and storage | 10 150 (0.09) | 3292 (0.05) | 8687 (0.07) | 4952 (0.07) |
| Unknown or private households | 91 (0.00) | 5 (0.00) | 61 (0.00) | 50 (0.00) |
| Water supply | 982 (0.01) | 222 (0.00) | 772 (0.01) | 560 (0.01) |
| Wholesale and retail | 24 783 (0.21) | 3132 (0.05) | 17 763 (0.15) | 10 964 (0.16) |
| **Company size** | | | | |
| 1–9 | 31 575 (0.26) | 3233 (0.05) | 22 050 (0.18) | 12 281 (0.18) |
| 10–49 | 48 126 (0.40) | 20 514 (0.30) | 43 030 (0.36) | 24 979 (0.36) |

Continued

**Table 1** Continued

|  | No IA, n (%) | IA, n (%) | No IA (IPTW) n (%) | IA (IPTW) n (%) |
|---|---|---|---|---|
| 50–249 | 12 555 (0.11) | 19 437 (0.28) | 20 198 (0.17) | 11 599 (0.17) |
| ≥250 | 27 107 (0.23) | 25 383 (0.37) | 33 142 (0.28) | 19 934 (0.29) |
| Region |  |  |  |  |
| East | 55 898 (0.47) | 29 625 (0.43) | 55 702 (0.47) | 31 407 (0.46) |
| Middle | 10 695 (0.09) | 6577 (0.10) | 10 405 (0.09) | 6180 (0.09) |
| North | 14 471 (0.12) | 7700 (0.11) | 14 002 (0.12) | 8849 (0.13) |
| South | 9459 (0.08) | 6399 (0.09) | 9773 (0.08) | 5692 (0.08) |
| West | 28 840 (0.24) | 18 266 (0.27) | 28 538 (0.24) | 16 665 (0.24) |
| Diagnosis |  |  |  |  |
| Musculoskeletal | 43 419 (0.36) | 21 570 (0.31) | 40 752 (0.34) | 24 199 (0.35) |
| Other types | 53 417 (0.45) | 34 490 (0.50) | 55 200 (0.46) | 32 385 (0.47) |
| Psychological | 22 527 (0.19) | 12 507 (0.18) | 22 467 (0.19) | 12 209 (0.18) |

in full-time). Furthermore, there was a clear reduction in non-employment of about 2 pp after 1 year for men with IA at the time the initial SA started.

Workers with musculoskeletal illnesses as cause for their initial SA, reached peak effect of IA on RTW after 27 days, where the probability was 3.7 pp (2.4 to 5.0) higher than without IA. This effect magnitude was relatively stable until around day 100, from where it declined and then fluctuated around 2 pp to the end of follow-up. The effect on work was mostly offset by reduced probability of SA, though a slight increase for graded SA was seen and a 2 pp lower probability of non-employment after 1 year.

In the group with psychological diagnoses, the effect on RTW reached a maximum after 37 days, with 5.6 pp (3.5 to 7.7) higher probability with IA. Compared with other subgroups, effects varied much more over time. From the

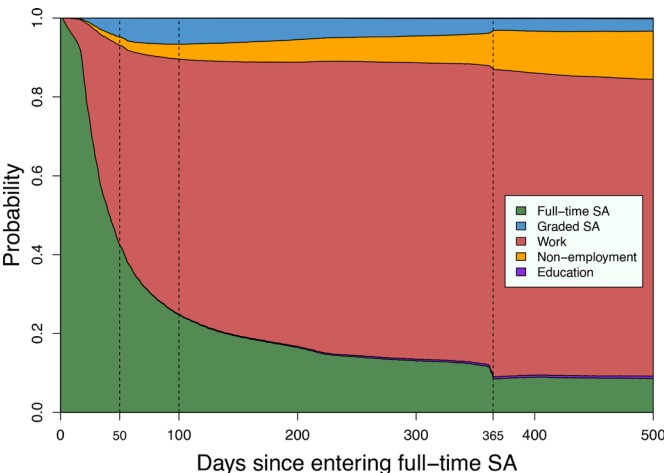

**Figure 2** Stacked probability plot of estimated unadjusted state probabilities, for the full study population (both intervention groups, n=187 930), after first initial entry into full-time sickness absence (SA). Individuals may have repeated spells of each state during follow-up, and the plot shows the probability of being in either of the states at all days during the follow-up period.

maximum effect of 5.6 pp, it dropped down to a 3 pp after 1 month, before increasing to over 5 pp 3 months later. The effect then varied between 3 and 5 pp before dropping down to 2 pp after 1 year. In the first year, the effect on RTW was mostly offset by reduced full-time SA, but a small increase in graded SA and reduction in non-employment were seen. One year after an initial SA episode due to psychological illnesses, the probability of non-employment was more than 2 pp lower with IA.

Average effects of IA versus no IA for subgroups are summarised in table 2 in terms of ELOS over the first year. For women, we found that IA led to on average 5.8 (95% CI 2.2 to 9.4) fewer days in full-time SA and 6.0 (95% CI 1.6 to 10.4) more days in work. ELOS in other states were not affected. In men, effects were larger. Men with IA could expect 9.4 (95% CI 5.5 to 13.3) fewer days in full-time SA and 11.2 (95% CI 6.4 to 16.1) more days in work. There was a small increase in ELOS for graded SA of 1.4 days (95% CI −0.4 to 3.2), but the CI included zero. Non-employment was lowered by 3.1 days (95% CI −5.8 to −0.5). For individuals with initial SA due to musculoskeletal illnesses, IA led to on average 8.3 (95% CI 3.7 to 12.9) fewer days in full-time SA and 8.9 (95% CI 3.3 to 14.5) more days in work. ELOS differences for graded SA and non-employment were small, and both confidence intervals included zero. Individuals with psychological diagnoses had the largest differences in ELOS. IA led to on average 14.1 (95% CI 6.0 to 22.3) more days in work over the first year and 11.1 (95% CI 4.2 to 18.0) fewer days of full-time SA. Non-employment was reduced by 3.7 days for the psychological diagnosis group, but the CI included zero.

### Sensitivity analyses

To study the impact of covariate adjustment and to investigate robustness of our results, we conducted a series of sensitivity analyses, as described earlier. The analyses supported our main findings, but also indicated effect differences between industries. In the sensitivity analysis

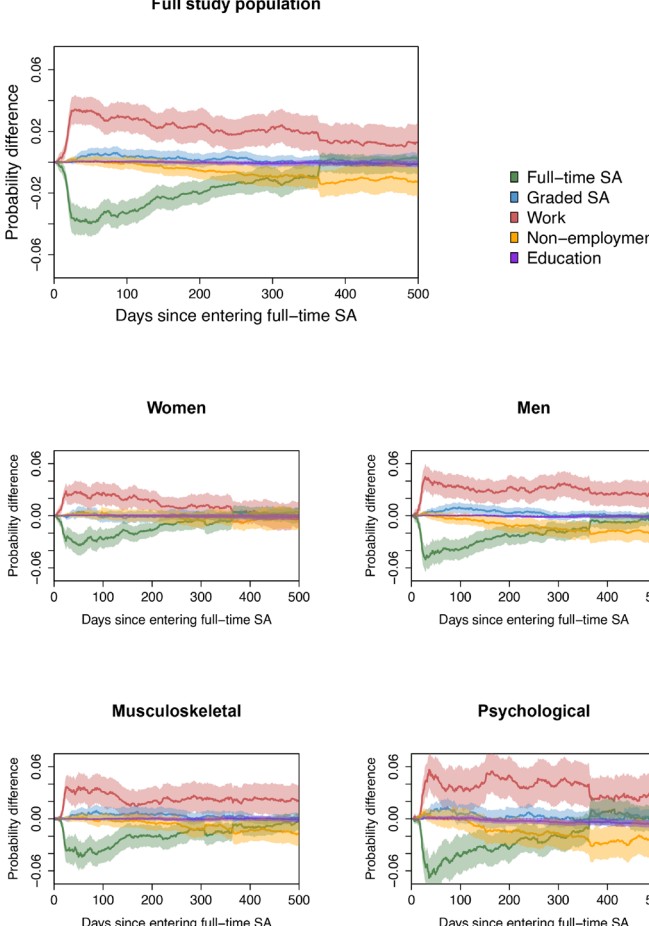

**Figure 3** Difference in estimated state probabilities, $\pi^1(t)-\pi^0(t)$, for 500 days following initial entry into full-time sickness absence (SA), where superscript 1 denotes having the inclusive working life agreement (IA) group and superscript 0 denotes not having IA. Estimates were adjusted for confounding using inverse probability of treatment (IPT) weighting. Faded areas are 95% CI based on 1000 bootstrap samples. The topmost plot in the figure shows results for the entire study population. The middle two plots are for women and men separately, and the bottom two are separated between musculoskeletal and psychological diagnoses pertaining to the initial SA spell.

of the woman in the study population, where all diagnoses related to pregnancy, childbearing and family planning were removed, we found a somewhat higher effect of IA than in the analysis which included all the women. Detailed results from these analyses are described in online supplemental sections S.2–S.6 and shown in online supplemental figures 1–5.

The sequence plots that illustrate individuals' multistate histories (see online supplemental figure 6) showed that the most common history was one with a short period of full-time SA, less than 2 months, followed by uninterrupted stay in the work state. We have provided more details on the sequence plots in online supplemental section S.7.

A barplot that shows gender representation in the largest industries is included in online supplemental

figure 7 and further described in online supplemental section S.8. Here we found that men worked mostly within construction, manufacturing and transport and women worked in education and health industries.

## DISCUSSION

We found that having access to the Norwegian IA Agreement, which included measures for preventing and reducing SA, at the time of initiating a first long-term SA episode, increased the probability of later work and decreased the probability of later full-time SA during the first 500 days after first entering SA. The probability of later non-employment was also reduced, particularly after 1 year. Larger effects were found for men than for women, and among individuals with musculoskeletal or psychological diagnoses connected to the initial SA spell. For women, the effect of IA was negligible after 1 year, while for men, the effect remained also after the first year.

A reduction in full-time SA and a slight increase in graded SA are in accordance with the first IA goal of reducing SA. In addition, it is worth noticing that non-employment 1 year after entering SA, appears to be reduced in workplaces with IA, especially for men. From a health perspective, the underlying intention of IA is to reduce SA by remaining in work, also after the first year when the right to certain SA benefits in Norway ceases. Our results show that having IA contributes to reducing SA by keeping people in work, rather than by existing work. Furthermore, while previous studies found that IA enterprises may have higher SA, or that IA has none or minor effects on the occurrence of SA,[8 11 12] our study shows that once on SA, IA contributed towards reducing the number of SA days. This could indicate that IA had a larger effect on SA duration than on the occurrence of SA.

A recent paper on the effect of IA on SA prevalence and duration, using similar data sources but other methods, found a beneficial effect of IA in terms of reduced duration for both musculoskeletal and psychological diagnoses, particularly in men.[12] Another study, from 2020, on IA and risk of long-term SA spells found small differences, but a slight decrease for female workers in stratified analyses.[11] Effects of IA are greater among individuals who had musculoskeletal or psychological diagnoses, compared with other diagnoses, is expected, as the intervention is specifically aimed at these patient groups.[6]

We estimated effects of having access to IA and did not investigate which IA measures had most effect or whether workers had accessed them. There is supporting evidence that various workplace health-promotion interventions may prolong working life in certain types of work.[25] A Cochrane review from 2015 found moderate evidence for workplace interventions being effective in increasing RTW in workers on sick leave with musculoskeletal disorders (first and lasting RTW) and mental health problems (only first RTW).[26] A systematic review from 2018[27] further supports these findings. However, a review of studies on

**Table 2** Expected length of stay (ELOS) in days, for 1 year after entering long-term full-time sickness absence (SA) from work, with 95% CI, based on 1000 bootstrap samples

| | IA | No IA | Difference |
|---|---|---|---|
| **Full study population** | | | |
| Full-time SA | 88.8 (87.2 to 90.5) | 96.4 (95.3 to 97.5) | −7.6 (−10.3 to −4.8)* |
| Graded SA | 18.1 (17.2 to 19) | 17.3 (16.7 to 17.9) | 0.8 (−0.7 to 2.2) |
| Work | 240.8 (238.7 to 242.8) | 232.4 (230.9 to 233.8) | 8.4 (4.9 to 11.9)* |
| Non-employment | 16.2 (15.1 to 17.3) | 17.8 (17.1 to 18.5) | −1.6 (−3.4 to 0.2) |
| Education | 0.9 (0.7 to 1.1) | 1.0 (0.8 to 1.1) | −0.1 (−0.5 to 0.3) |
| **Women** | | | |
| Full-time SA | 94.2 (92.3 to 96.1) | 100.0 (98.3 to 101.7) | −5.8 (−9.4 to −2.2)* |
| Graded SA | 21.9 (20.7 to 23.2) | 21.9 (21.1 to 22.8) | 0.0 (−2.1 to 2.1) |
| Work | 232.3 (230.0 to 234.6) | 226.3 (224.2 to 228.4) | 6.0 (1.6 to 10.4)* |
| Non-employment | 15.3 (14.0 to 16.6) | 15.5 (14.4 to 16.5) | −0.2 (−2.5 to 2.1) |
| Education | 1.2 (0.8 to 1.6) | 1.2 (1.0 to 1.5) | 0.0 (−0.6 to 0.6) |
| **Men** | | | |
| Full-time SA | 82.9 (80.3 to 85.5) | 92.3 (91.0 to 93.6) | −9.4 (−13.3 to −5.5)* |
| Graded SA | 13.4 (12.2 to 14.7) | 12.0 (11.4 to 12.6) | 1.4 (−0.4 to 3.2) |
| Work | 250.3 (247.2 to 253.5) | 239.1 (237.4 to 240.8) | 11.2 (6.4 to 16.1)* |
| Non-employment | 17.5 (15.6 to 19.3) | 20.6 (19.8 to 21.4) | −3.1 (−5.8 to −0.5)* |
| Education | 0.6 (0.3 to 0.8) | 0.7 (0.6 to 0.9) | −0.1 (−0.6 to 0.3) |
| **Musculoskeletal** | | | |
| Full-time SA | 92.0 (89.2 to 94.8) | 100.3 (98.4 to 102.1) | −8.3 (−12.9 to −3.7)* |
| Graded SA | 20.2 (18.7 to 21.8) | 18.7 (17.9–19.6) | 1.5 (−1.0 to 3.9) |
| Work | 236.0 (232.4 to 239.6) | 227.5 (225.2 to 229.8) | 8.5 (2.6 to 14.4)* |
| Non-employment | 16.1 (14.3 to 17.8) | 17.7 (16.7 to 18.6) | −1.6 (−4.3 to 1.1) |
| Education | 0.7 (0.4 to 1.1) | 0.8 (0.5 to 1) | 0.0 (−0.6 to 0.5) |
| **Psychological** | | | |
| Full-time SA | 104.3 (100.1 to 108.6) | 115.4 (112.7 to 118.1) | −11.1 (−18.0 to −4.2)* |
| Graded SA | 22.4 (20.2 to 24.5) | 21.2 (19.9 to 22.5) | 1.2 (−2.2 to 4.6) |
| Work | 213.0 (207.9 to 218.1) | 198.9 (195.9 to 201.9) | 14.1 (6.0 to 22.3)* |
| Non-employment | 23.9 (20.7 to 27.1) | 27.7 (26.0 to 29.3) | −3.7 (−8.6 to 1.2) |
| Education | 1.1 (0.6 to 1.7) | 1.7 (1.3 to 2.2) | −0.6 (−1.6 to 0.4) |

*The last column shows the difference in ELOS between IA and no IA, where an asterisk indicates CIs of the difference that do not include zero.
IA, Agreement for a More Inclusive Working Life.

the effect of interventions for improving RTW in workers with common mental illness found that the available interventions did not lead to improved RTW rates.[28] One should also note, as IA has been gradually introduced at a national level in Norway over the study period and beyond, that it is likely that IA has had an impact on SA and RTW at a societal level. This effect is not picked up in our study, as we consider individual differences of having IA or not over the study period.

A strength of this study is the use of a large cohort with high completeness, with detailed information on individual longitudinal work participation and a large set of confounders. Another strength is the use of multistate

modelling. We believe that the study serves as a good example of how longitudinal data from multiple registries can be used to construct detailed outcome trajectories and that IPT weighted multistate models offer a suitable way of estimating effects on such outcomes. The approach allows us to analyse more detailed long-term work-related outcomes than traditional approaches, and to calculate various relevant outcome measures, ELOS being one example. However, there are also various limitations. Most importantly, this is an observational study, and access to the measures given through IA was not randomised, but subject to the decision of the company signing up for IA. Even though adjustment for

various assumed confounding variables was made, there is no guarantee that residual confounding is not present. However, sensitivity analyses were not able to detect residual confounding in settings where more detailed confounder information was available. The negative control by analysing pre-exposure multistate histories indicated the same. Note that if the negative control had showed different pre-exposure histories, even after initial confounder adjustment, these histories could also be adjusted for as additional baseline information.

Given the assumptions of no unmeasured confounding, positivity and consistency, the IPT weighted multistate model aims to mimic the scenario one would see if access to IA was randomised at baseline (time of first long-term SA episode). Note also that we implicitly assume that IA exposure acts as if given at first SA. As our analyses are restricted to individuals without any recent SA history, we believe this to be a reasonable assumption. The mechanisms through which IA potentially promotes faster RTW are many. One can assume that some are connected to individual measures undertaken at IA workplaces, while others are related to measures available for IA workplaces in general.

There were notable differences in the estimated effects of IA for women and men, which can have various explanations. The Norwegian labour market has a high level of gender segregation,[29 30] which can also be observed in our study population. For our study sample, a high proportion of men worked in construction, manufacturing and transport, while women dominated the educational and healthcare industries. Although the available measures are the same for every IA company, it is reasonable to expect that the implementation and effect of IA measures vary substantially across workplaces, occupations and industries. The governmental appointed Research Group for the IA Agreement writes in their report from 2018[31] that there are substantial differences between industries and types of work regarding challenges with SA and SA reducing measures. Also, as observed in most western countries over many years,[32–34] women generally have a higher prevalence of SA than men. The gender differences in labour outcomes are the subject of extensive research and debate. Some studies have indicated that differences in work-related outcomes can be caused by unfavourable working conditions among women,[35 36] while others find less favourable conditions among men.[33] Some studies suggest women handle work-related strain worse than men.[37] There are also results indicating that the double burden of household work affects women to a higher degree than men,[38] and if the SA comes as a result of challenges at home, measures focusing on the workplace might not be as effective. A qualitative study of interventions and rehabilitation activities in connection with RTW from 2021[39] found that women expressed a need for home-related interventions, whereas men expressed a need for organisational interventions to counter feelings of resignation at work. As the IA Agreement is focused on workplace-related measures, it may therefore be the case

that the IA measures better target men. Furthermore, a substantial part of the initial SA cases among women in our study population was in connection with pregnancy, childbearing and family planning, and IA measures may not be as relevant for SA tied to such diagnoses. Online supplemental analysis showed that when excluding these cases, the effects of IA among the remaining women were somewhat higher.

The IA Agreement is specific to Norway, and it is not likely that similar effects would have been seen in other countries. Norway is known for having a generous welfare system and high protection for individual workers. In addition, the cohort studied was young (28–43 years), and results do not necessarily generalise to older populations. Future work should focus on studying specific measures for preventing and reducing SA and how they vary with different types of work, industries, diagnosis types and other worker characteristics.

**Author affiliations**
[1]Oslo Centre for Biostatistics and Epidemiology, Oslo University Hospital, Oslo, Norway
[2]Oslo Centre for Biostatistics and Epidemiology, Department of Biostatistics, University of Oslo, Oslo, Norway
[3]Research Group for Occupational Medicine and Epidemiology, National Institute of Occupational Health, Oslo, Norway
[4]Research Group for Work Psychology and Physiology, National Institute of Occupational Health, Oslo, Norway
[5]Department of Community Medicine and Global Health, University of Oslo, Oslo, Norway

**Acknowledgements** The late Professor Emeritus Tor Bjerkedal (1926–2015) conceived the idea for and took part in establishing this registry-based cohort in 2002.

**Contributors** RH, NM and JMG conceptualised the study. NM, RLH, SLM, KU, PK, ISM and JMG contributed to the interpretation of the data and study design. RH and NM performed the data preparation and data analysis with the contribution of JMG. All authors contributed to the interpretation of the results. RH prepared the first draft of the manuscript. All authors commented on previous versions, contributed to revised versions and approved the final manuscript. RH acts as the guarantor for the overall content and attests that all listed authors meet authorship criteria and that no others meeting the criteria have been omitted.

**Funding** This work was supported by the Research Council of Norway (Grant number 273674).

**Competing interests** None declared.

**Patient and public involvement** Patients and/or the public were not involved in the design, or conduct, or reporting, or dissemination plans of this research.

**Patient consent for publication** Not applicable.

**Ethics approval** This study involves human participants and was approved by Regional Committees for Medical and Health Research Ethics (REC) licence number 17344. This research study was conducted retrospectively from data provided by third party national Norwegian registries and the use of the data was sanctioned under Regional Committees for Medical and Health Research Ethics (REC) licence number 17344. Informed consent is not needed for this registry-based study according to Norwegian law (The Personal Data Act, Section 8 and 9, and The General Data Protection Regulation Article 6.1 e) and Article. 9.2 j)).

**Provenance and peer review** Not commissioned; externally peer reviewed.

**Data availability statement** Data may be obtained from a third party and are not publicly available. The data underlying this article are considered sensitive and subject to strict regulation and cannot be shared publicly due to privacy of individuals included in the study.

peer-reviewed. Any opinions or recommendations discussed are solely those of the author(s) and are not endorsed by BMJ. BMJ disclaims all liability and responsibility arising from any reliance placed on the content. Where the content includes any translated material, BMJ does not warrant the accuracy and reliability of the translations (including but not limited to local regulations, clinical guidelines, terminology, drug names and drug dosages), and is not responsible for any error and/or omissions arising from translation and adaptation or otherwise.

**ORCID iDs**
Rune Hoff http://orcid.org/0000-0002-7695-8165
Petter Kristensen http://orcid.org/0000-0003-0469-5033

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
