## [Reviewer comments · BMJ Open]

ARTICLE DETAILS

TITLE (PROVISIONAL)	An initiative for a more inclusive working life and its effect on return-to-work after sickness absence – a multi-state longitudinal cohort study
AUTHORS	Hoff, Rune; Maltzahn, Niklas; Hasting, Rachel; Merkus, Suzanne L; Undem, Karina; Kristensen, Petter; Mehlum, Ingrid; Gran, Jon

VERSION 1 – REVIEW

REVIEWER	Serra, Laura Universitat de Girona Facultat de Ciències Econòmiques i Empresarials, GRECS-Research Group on Statistics, Econometrics and Health
REVIEW RETURNED	29-Apr-2022

GENERAL COMMENTS	This study examines the effect of the tripartite Agreement for a More Inclusive Working Life (IA) established in Norway in 2001. Using observational longitudinal registry data, the authors use multi-state modelling and the effect of IA is assessed by estimating differences in state probabilities over time using inverse probability of treatment (IPT) weighting. In general, the findings suggest that IA increases the probability of work after SA and that these results are higher in men than in women. Similar trends are found also within musculoskeletal and psychological diagnoses. In my opinion, the paper is well written and the methodology is well applied to achieve the objectives of the study. However, I have some comments that I think could improve the manuscript in its current form. Main comments I am not very familiar with the methodology used by the authors and I imagine that the IPT weighted takes into account the differences observed regarding the application of the IA. Companies with IA are, in general, big companies and they include a larger number of industries of education and health and social work. In contrast, those without IA are more represented by construction and wholesale and retail industries. These characteristics could influence the results and I consider that these differences should be noticed by the authors. In addition, workers' health is very related to their jobs and so, probably workers in the construction have worst health on the baseline than those working in the field of education. I think that this is something that authors should be aware of. I would suggest that the authors include an additional analysis based on the sequence analysis. This analysis would describe the transition that workers do among the different states represented in Figure 1 which could be very interesting to better understand the
--

	pattern of these workers during these 500 days of follow up (https://cran.r-project.org/web/packages/TraMineR/TraMineR.pdf) Finally, I have a concern regarding the relapses. What happens if the worker that you are following up have another episode of SA? Do you finish following him/her or do you continue their follow up as if it was the first episode? I think that this should be clarified. Major issues Data Sources and data preparation Section. The paper is light on important details about the data and the construction of its variables. I would recommend expanding this section which will directly make clearer the application of the methodology. Discussion Section. I miss a larger explanation regarding the differences observed between men and women. The authors give a very brief argument. I think the authors have to work harder on these ideas and try to contrast their results to others found in the literature. In general, in my opinion, this section needs more effort to discuss your results against constructively the existing literature. Minor issues Figure 2: I would recommend including some vertical lines to clarify the percentages explained in the text. Table 2: I would recommend including a mark, such as an asterisk, to indicate the significance of the variable. References. I would recommend updating some of the references as they are a little bit ancient. So, I would suggest publishing this manuscript but only after carefully addressing all the points presented above.
--	---

REVIEWER	Langhammer, Birgitta Oslo Metropolitan University, Department of Physiotherapy
REVIEW RETURNED	12-May-2022

GENERAL COMMENTS	Comments BMJ open: « An initiative for a more inclusive working life and its effect on RTW after sickness absence- a multistate longitudinal cohort Title: what exactly is meant with multi-state? It is only Norway that is at focus? ok definition on p 4 Introduction: ok Methods: ok Results: ok Discussion: ok, including limitations Conclusion: ok Tables: ok Figures: ok References: ok
--

VERSION 1 – AUTHOR RESPONSE

Reviewer: 1

Dr. Laura Serra, Universitat de Girona Facultat de Ciències Econòmiques i Empresarials

Comments to the Author:

This study examines the effect of the tripartite Agreement for a More Inclusive Working Life (IA) established in Norway in 2001. Using observational longitudinal registry data, the authors use multi-state modelling and the effect of IA is assessed by estimating differences in state probabilities over time using inverse probability of treatment (IPT) weighting.

In general, the findings suggest that IA increases the probability of work after SA and that these results are higher in men than in women. Similar trends are found also within musculoskeletal and psychological diagnoses.

In my opinion, the paper is well written and the methodology is well applied to achieve the objectives of the study. However, I have some comments that I think could improve the manuscript in its current form.

Main comments:

I am not very familiar with the methodology used by the authors and I imagine that the IPT weighted takes into account the differences observed regarding the application of the IA. Companies with IA are, in general, big companies and they include a larger number of industries of education and health and social work. In contrast, those without IA are more represented by construction and wholesale and retail industries. These characteristics could influence the results and I consider that these differences should be noticed by the authors. In addition, workers' health is very related to their jobs and so, probably workers in the construction have worst health on the baseline than those working in the field of education. I think that this is something that authors should be aware of.

Response: *Thank you for this comment. We agree and adjusting for these differences (using the observed confounders) is the purpose of the inverse probability weighting. To make this clearer, we have now explained this more explicitly in second paragraph of the "Outcome and confounders" section and the "Estimands and statistical methods" section. We have also noted the imbalance in characteristics which is visible before weighting, and the balance we achieve after weighting, which is visible in Table 1, in the "Descriptive Statistics" section under "Results". We have also expanded the description of the multi-state modelling under "statistical methods" to make this part clearer, with references added for further reading.*

I would suggest that the authors include an additional analysis based on the sequence analysis. This analysis would describe the transition that workers do among the different states represented in Figure 1 which could be very interesting to better understand the pattern of these workers during these 500 days of follow up (<https://cran.r-project.org/web/packages/TraMineR/TraMineR.pdf>)

Response: *Thank you for this suggestion. We have now included sequence plots of 5 main industries separated by IA-status in the supplementary material and added a note on this in the "Subgroup and sensitivity analyses" section.*

Finally, I have a concern regarding the relapses. What happens if the worker that you are following up have another episode of SA? Do you finish following him/her or do you continue their follow up as if it was the first episode? I think that this should be clarified.

Response: All individuals are followed for 500 days, and the multi-state model considers any number of relapses and movement between states. To make this (key) point clearer, we have elaborated on this in the “Outcomes and confounders” section and in the first paragraph of the “Estimands and statistical methods” section. We have also modified the figure texts to help clarify this.

Major issues:

Data Sources and data preparation Section. The paper is light on important details about the data and the construction of its variables. I would recommend expanding this section which will directly make clearer the application of the methodology.

Response: Thank you for this suggestion. The “Data sources and data preparation” section has now been expanded and largely re-written to make this part much clearer.

Discussion Section. I miss a larger explanation regarding the differences observed between men and women. The authors give a very brief argument. I think the authors have to work harder on these ideas and try to contrast their results to others found in the literature.

Response: Thank you pointing out this. We agree that the initial discussion of gender differences was short and have now substantially expanded this part of the “Discussion” section. We have also done an additional supplementary analysis, excluding woman with initial SA related to pregnancy, childbearing and family planning, to shed more light on the topic.

In general, in my opinion, this section needs more effort to discuss your results against constructively the existing literature.

Response: The expanded discussion now includes a more comprehensive comparison to existing literature.

Minor issues:

Figure 2: I would recommend including some vertical lines to clarify the percentages explained in the text.

Response: *Vertical lines are now added on day 50, 100 and 365.*

Table 2: I would recommend including a mark, such as an asterisk, to indicate the significance of the variable.

Response: *Asterisks are added in the last column to highlight the confidence intervals for difference in ELOS not covering zero.*

References. I would recommend updating some of the references as they are a little bit ancient

Response: *We have now removed the most ancient one, and added several from more recent years.*